**Data Availability Statement:** All relevant data are within the manuscript and its Supporting Information.

**Funding:** The authors received no specific funding for this work.

# Megacystis in the first trimester of pregnancy: Prognostic factors and perinatal outcomes

**Emmanuelle Lesieur**[1]*, **Mathilde Barrois**[1], **Mathilde Bourdon**[2,3,4], **Julie Blanc**[5,6], **Laurence Loeuillet**[7], **Clémence Delteil**[8,9], **Julia Torrents**[10], **Florence Bretelle**[5,11], **Gilles Grangé**[1], **Vassilis Tsatsaris**[1,12], **Olivia Anselem**[1]

1 Maternité Port-Royal, AP-HP, Hôpital Cochin, FHU PREMA, Paris, France, 2 Faculté de Médecine Paris Centre, Faculté de Santé, Université de Paris, Paris, France, 3 Service de Gynécologie-Obstétrique II et de Médecine de la Reproduction, AP-HP, Centre Hospitalier Universitaire (CHU) Cochin, Paris, France, 4 Department "Infection, Immunity and Inflammation", Université de Paris, Institut Cochin, Paris, France, 5 Service de Gynécologie Obstétrique, Hôpital Nord, AP-HM, Chemin des Bourrely, Marseille, France, 6 EA3279, CEReSS, Health Service Research and Quality of Life Center, Université Aix-Marseille, Marseille, France, 7 Service d'Histologie-Embryologie-Cytogénétique, Hôpital Necker-Enfants Malades, AP-HP, Paris, France, 8 Institut Médicolégal de Marseille, Hôpital Timone Adultes, Marseille, France, 9 CNRS, EFS, ADES UMR 7268, Aix-Marseille université, Marseille, France, 10 Service d'Anatomo-Cytopathologie et Fœtopathologie, Hôpital de la Timone, Marseille, France, 11 Aix Marseille Univ, IRD, AP-HM, MEΦI, Marseille, France, 12 Université de Paris, Inserm UMR-S 1139, Physiopathologie et Pharmacotoxicologie Placentaire Humaine, Paris, France

* Emmanuelle.lesieur@ap-hm.fr

## Abstract

### Objective

To determine whether bladder size is associated with an unfavorable neonatal outcome, in the case of first-trimester megacystis.

### Materials and methods

This was a retrospective observational study between 2009 and 2019 in two prenatal diagnosis centers. The inclusion criterion was an enlarged bladder (> 7 mm) diagnosed at the first ultrasound exam between 11 and $13^{+6}$ weeks of gestation. The main study endpoint was neonatal outcome based on bladder size. An adverse outcome was defined by the completion of a medical termination of pregnancy, the occurrence of in utero fetal death, or a neonatal death. Neonatal survival was considered as a favorable outcome and was defined by a live birth, with or without normal renal function, and with a normal karyotype.

### Results

Among 75 cases of first-trimester megacystis referred to prenatal diagnosis centers and included, there were 63 (84%) adverse outcomes and 12 (16%) live births. Fetuses with a bladder diameter of less than 12.5 mm may have a favorable outcome, with or without urological problems, with a high sensitivity (83.3%) and specificity (87.3%), area under the ROC curve = 0.93, 95% CI (0.86–0.99), p< 0.001. Fetal autopsy was performed in 52 (82.5%) cases of adverse outcome. In the 12 cases of favorable outcome, pediatric follow-up was normal and non-pathological in 8 (66.7%).

**Competing interests:** The authors have declared that no competing interests exist.

## Conclusion

Bladder diameter appears to be a predictive marker for neonatal outcome. Fetuses with smaller megacystis (7–10 mm) have a significantly higher chance of progressing to a favorable outcome. Urethral stenosis and atresia are the main diagnoses made when first-trimester megacystis is observed. Karyotyping is important regardless of bladder diameter.

## Introduction

First-trimester ultrasound is a fundamental element of screening policy. Over the past few decades, technical improvements in ultrasound equipment have ameliorated understanding and visualization of fetal anatomy in the first trimester. Thus, first-trimester ultrasound seems ready to evolve from a simple screening examination to a detailed anatomical examination traditionally performed in the second trimester of pregnancy [1–5].

Ultrasound, which is complementary to noninvasive prenatal testing [6], offers multiple advantages, including earlier and more precise diagnosis, despite an uncertain prognosis at this term of pregnancy [7–9].

Megacystis in the first trimester of pregnancy is usually defined by a bladder size > 7 mm between 11 and $13^{+6}$ weeks of gestation [10–12], after checking for bladder emptying during the exam [13]. It occurs in 1/1600 to 1/3000 pregnancies. The cause may be obstructive in 60% of cases (posterior urethral valves, urethral atresia or urethral stenosis, cloacal anomalies), non-obstructive in 30% of cases, mainly syndromic disease (megacystis-microcolon-intestinal hypoperistalsis syndrome, prune belly syndrome), and finally idiopathic or transient (10% of cases) [14].

Several authors have evaluated the prognostic factors and neonatal outcomes associated with megacystis, all terms of pregnancy combined [15–21], as well as the possibility of antenatal surgical intervention (vesicoamniotic shunt or ablation of the obstructive tissue through in utero cystoscopy) in obstructive megacystis [22–27]. However, few authors have specifically evaluated the prognosis associated with bladder size and etiology [12, 28–30], and in most cases these were limited series. The main objective of this work was to evaluate in first-trimester megacystis whether bladder size is associated with unfavorable outcome. The secondary endpoint was description of etiologies and evaluation of whether there was an association between bladder size and etiology.

## Materials and methods

### Data collection and population studied

This was a retrospective observational study performed between November 2009 and November 2019 in two prenatal diagnosis centers: the Cochin Hospital, Paris, France and the Timone Hospital, Marseille, France. All patients were informed during ultrasound examinations of the possible use of their data for scientific purposes and could at any time indicate their refusal. Written consent was given by adults as well as by parents for minors and by guardians for persons under guardianship. A request for processing of this data was made to the Committee of Patient Data Protection and was accepted (French PADS). All data have been fully anonymized.

Fetal megacystis was defined as a longitudinal bladder diameter ≥ 7 mm measured from the bladder dome to the bladder neck in the midsagittal plane on an ultrasound scan performed between 11 and $13^{+6}$ weeks of gestation and confirmed by an ultrasound specialist. We excluded any diagnosis of megacystis beyond this term.

Bladder diameter was investigated with respect to the likelihood of postnatal outcome (favorable with or without urological problems or adverse outcome), respectively. Sensitivity, specificity and area under the receiver operating characteristic (ROC) curve (AUC) were calculated.

Based on the ROC curve, two groups were compared: (i) bladder diameter under 12.5 mm; (ii) bladder diameter greater than 12.5 mm. Our primary study endpoint was neonatal outcome (adverse or favorable) based on bladder size. An adverse outcome was defined by the completion of a medical termination of pregnancy (TOP), the occurrence of in utero fetal death (IUFD), or a neonatal death (ND). In this work, neonatal survival was considered as a favorable outcome and was defined by a live birth, with or without normal renal function, and with a normal karyotype. Medical data for pediatric follow-up of liveborn children were checked during their first year of life.

Our secondary endpoint was description of etiologies and evaluation of whether there was an association between bladder size and the different etiologies of megacystis (fetal karyotyping and pathological examination of the fetus).

All patients underwent detailed sonographic examination twice: a first ultrasound examination and another ultrasound examination one week later, to assess progression. Patients were included when the longitudinal bladder diameter at the first ultrasound was $\geq$ 7 mm. When the parents wished to continue the pregnancy, close ultrasound follow-up was offered. Nuchal translucency, bladder diameter, appearance of kidneys (normal, hydronephrosis, abnormal renal cortical appearance), amount of amniotic fluid and presence of associated elements (ascites, pulmonary hypoplasia) were systematically reported whatever the pregnancy outcome. Fetal karyotyping by chorionic villus sampling was systematically offered to the patients after ultrasound diagnosis in the two prenatal diagnosis centers. In the case of TOP or intrauterine death, a fetal autopsy was suggested (vacuum system or vaginal delivery). Vaginal delivery was encouraged to facilitate the pathological examination.

The final (pathological) diagnosis was divided into two categories:

- First, the obstructive causes (lower urinary tract obstruction diagnosis or LUTO diagnosis) encompassing the diagnosis of posterior urethral valve (PUV), urethral atresia, urethral stenosis, cloacal anomalies and ureterocele.

- Second, "other diagnoses", including syndromic disease (prune belly syndrome, megacystis-microcolon-intestinal hypoperistalsis syndrome) and other diagnoses not corresponding to obstructive causes. When pregnancy was continued, neonatal outcomes were obtained from medical records and pediatric care [31–34].

## Statistical analysis

Statistical analyses were performed using IBM SPSS Statistics software, version 20.0 (SPSS Inc., IL, USA). Continuous variables are presented as a mean with standard deviation range. Categorical variables are presented as numbers and percentages. Antenatal characteristics and postnatal outcome were then compared according to bladder size at the first ultrasound. We used the Pearson's $X^2$ test for comparison of qualitative variables. The area under the ROC curve was used to define a cut-off value. The statistical significance was defined as $p < 0.05$.

## Results

### Analysis of the population studied

Seventy-five cases of first-trimester fetal megacystis at the first ultrasound were included in our study population. Sixty-three percent of patients opted for TOP (n = 47/75), 14.7% of fetuses died

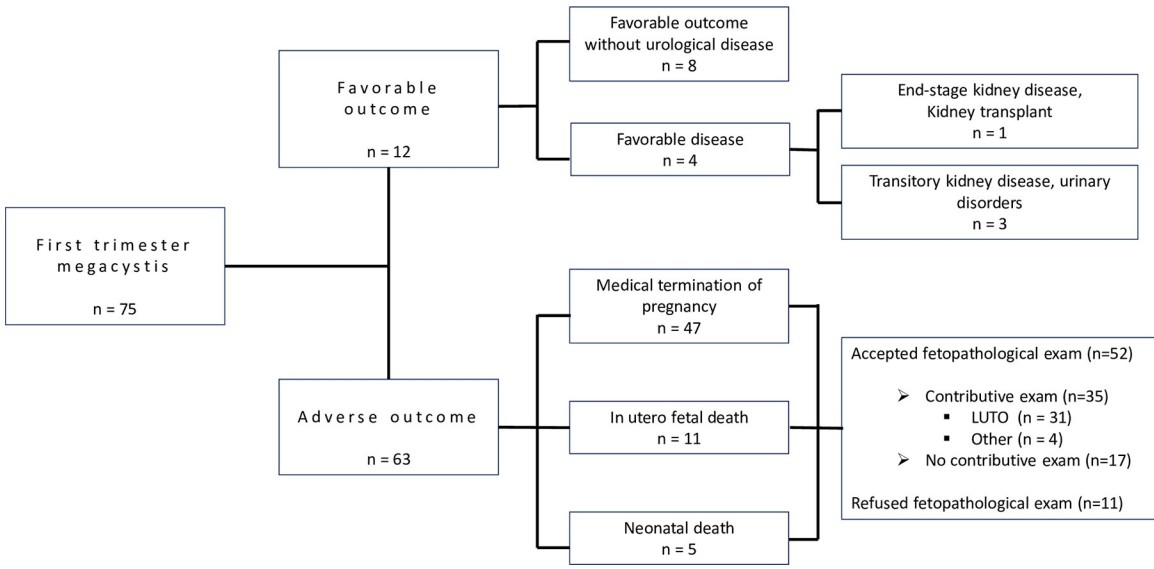

**Fig 1. Flow chart.**

in utero (n = 11/75) and 6.7% postnatally (5/75). There were 16% live births (n = 12/75). Therefore, we report 63 adverse outcomes (84%) and 12 favorable outcomes (16%). (Fig 1).

Population characteristics are detailed in Table 1. The median maternal age at diagnosis was 31 years (+/- 5.69). The median gestational age at diagnosis was 12.5 weeks of gestation

**Table 1. Population studied.**

| Population characteristics (n = 75) | |
| --- | --- |
| **Median maternal age (years)** | 31 |
| **Median gestational age at diagnosis (WG)** | 12.5 |
| **First ultrasound** | |
| **Nuchal translucency** | |
| < 95th percentile | 70 (93.3%) |
| >95th percentile | 5 (6.7%) |
| **Bladder size** | |
| < 12.5 mm | 14 (18.6%) |
| >12.5 mm | 61 (81.4%) |
| **Kidneys** | |
| Normal appearance | 48 (64%) |
| Abnormal renal cortical appearance | 11 (14.7%) |
| Hydronephrosis | 21 (28%) |
| **Amount of amniotic fluid** | |
| Normal quantity | 70 (93.3%) |
| Oligohydramnios | 5 (6.7%) |
| **Fetal gender** | |
| Male | 57 (76%) |
| Female | 18 (24%) |
| **Outcome** | |
| Favorable | 12 (16%) |
| Adverse | 63 (84%) |

(+/-1.79). The population consisted mainly of male fetuses (76%). Nuchal translucency was less than the 95th percentile in 69 cases (92%). During the first-trimester ultrasound examination, 64% of fetuses had kidneys of normal appearance (n = 48) and 90% had a normal amount of amniotic fluid (68%) (Table 1).

**Fetal outcome according to bladder size.** A ROC curve of bladder diameter was used to identify the optimal "cut-off" for prediction of neonatal outcome of megacystis in the first trimester (AUC = 0.93, [95% CI 0,86–0.99], p< 0.001). Based on the ROC curve analysis, the optimal "cut-off" bladder diameter was 12.5 mm. Under this cut-off, the probability of a favorable outcome was 91%, with 83.3% sensitivity (95% CI 33.3–91.7) and 87.3% specificity (95% CI 75–99.9) (Fig 2).

**Fetal characteristics according to bladder size.** For this analysis, patients were allocated to two groups, according to the size of the fetal bladder determined by the ROC curve: 14 fetuses (18.6%) had a bladder size under 12.5 mm mm, 61 fetuses (81.4%) had a bladder size greater than 15 mm. Fetal characteristics for each group are detailed in Table 2.

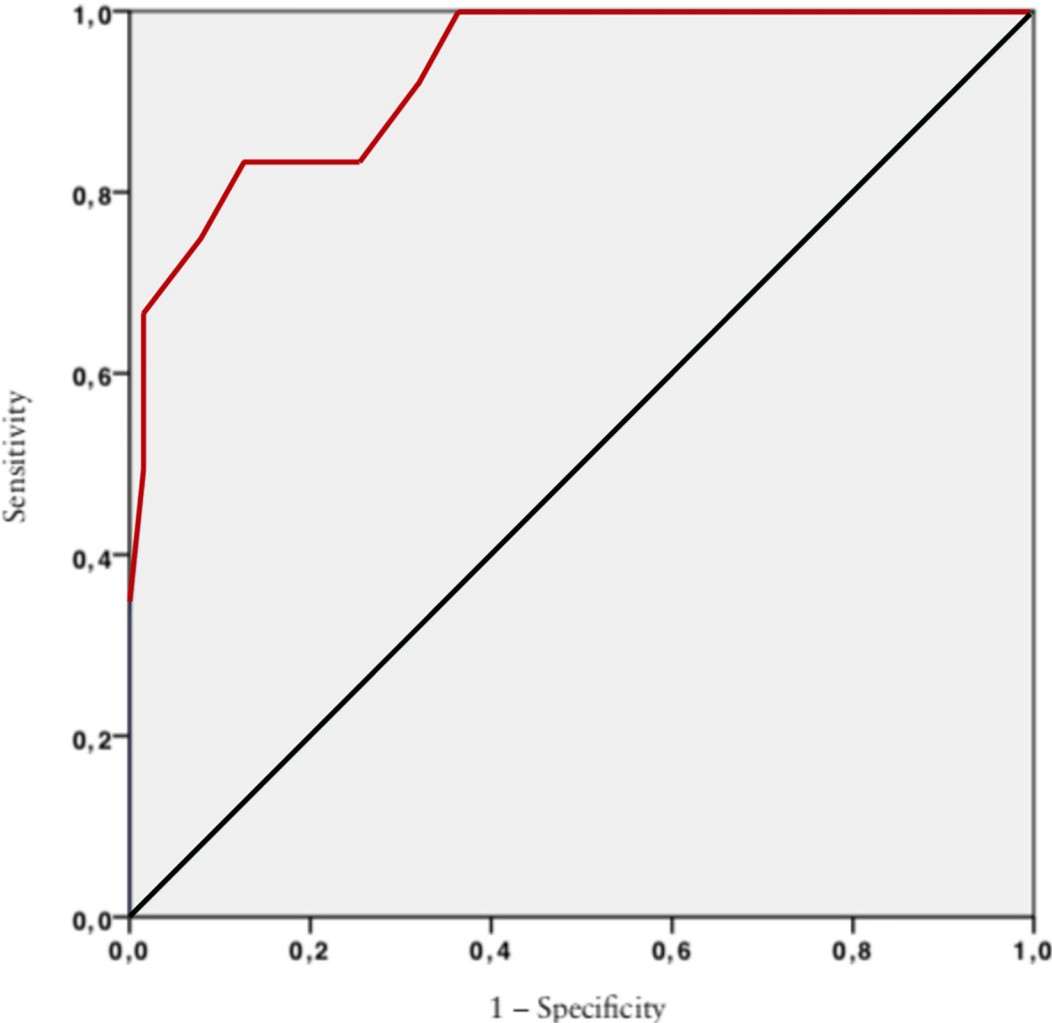

**Fig 2. Prediction of neonatal outcome: Performance of bladder diameter in the prediction of favorable or adverse outcome of megacystis in the first trimester.**

**Table 2. Univariate analysis: Analysis of US antenatal characteristics and neonatal outcomes according to bladder size.**

| Antenatal characteristics and postnatal outcome | Bladder size | | p |
|---|---|---|---|
| | < 12.5 mm | >12.5 mm | |
| | n = 14 (18.6%) | n = 61 (81.4%) | |
| **Age (years)** | | | |
| < 30 | 5 (35.7) | 21 (33.4) | 0.81 |
| 31–40 | 9 (64.3) | 32 (52.4) | |
| >40 | 0 | 8 (13.1) | |
| **Nuchal translucency** | | | |
| < 1.5 | 8 (57.1) | 25 (40.9) | 0.17 |
| 1.5–3 | 6 (42.8) | 31 (50.8) | |
| >3 | 0 | 5 (8.3) | |
| **Kidney appearance (FU)** | | | |
| Normal appearance | 11 (78.5) | 32 (52.4) | **0.02** |
| Hydronephrosis | 3 (21.4) | 18 (29.5) | **0.01** |
| Abnormal cortical appearance | 0 | 11 (18) | 0.26 |
| **Amount of amniotic fluid (FU)** | | | |
| Normal | 11 (78.5) | 56 (91.8) | 0.93 |
| Oligohydramnios/anhydramnios | 3 (21.4) | 5 (8.2) | 0.71 |
| **Kidney Appearance (SU)** | | | |
| Normal aspect | 8 (57.1) | 4 (6.5) | 0.09 |
| Hydronephrosis | 3 (21.4) | 33 (54.1) | **0.01** |
| Abnormal cortical aspect | 1 (7.1) | 40 (65.6) | **0.01** |
| **Amount of amniotic fluid (SU)** | | | |
| Normal | 12 (85.7) | 19 (31.1) | 0,12 |
| Oligohydramnios/anhydramnios | 2 (14.3) | 42 (68.8) | **<0.001** |
| **Fetal gender** | | | |
| Female (XX) | 4 (28.5) | 14 (22.9) | 0.47 |
| Male (XY) | 10 (71.4) | 47 (77.1) | |
| **Outcome (TOP excluded)** | | | |
| Favorable outcome | 12 (100) | 0 | **< 0.001** |
| IUFD | 0 | 11 | |
| ND | 0 | 5 | |
| **Outcome (TOP included)** | | | |
| IUFD | 0 | 11 (18) | **< 0.001** |
| ND | 0 | 5 | |
| TOP | 2 (14.2) | 50 (82) | |
| Favorable outcome | 12 (85.8) | 0 | |

(FU: first ultrasound; SU: second ultrasound; TOP: termination of pregnancy; IUFD: in utero fetal death; ND: neonatal death). The Chi-square test was used for statistical analysis.

There was no statistical difference between the groups concerning the mother's age, nuchal translucency or fetal gender.

In groups of fetuses with bladder size > 12.5 mm, the rate of hydronephrosis and oligohydramnios at the first-trimester ultrasound was significantly higher (p = 0.01). The rate of oligohydramnios was significantly higher too in fetuses with a bladder size > 12.5 mm (p<0.001), compared to fetuses with smaller bladder size.

Neonatal outcomes are described in Table 2. The rate of IUFD or ND was significantly higher in fetuses with a larger bladder diameter, whether or not TOP is taken into account in the statistical analysis.

**Antenatal and postnatal characteristics of fetuses according to neonatal outcome.** All infants with a favorable outcome (n = 12) had a normal bladder size at ultrasound at one-week control (spontaneous regression of megacystis). Spontaneous regression was observed in all cases when bladder size was < 12.5 mm at the first-trimester ultrasound scan. Renal appearance was normal, as was the amount of amniotic fluid in the ultrasound control one week later and on all other ultrasound scans during pregnancy. In the case of spontaneous regression, there was a favorable outcome without urological disease at one year.

Of the 12 children with a favorable outcome, 8 (66.7%) had normal renal function and non-pathological pediatric follow-up. Three children had transitory kidney disease (abnormal transitory renal function) and urinary disorders and needed pediatric urological monitoring. One child had an end-stage kidney disease at birth (abnormal renal function) that required a kidney transplant (Fig 1). Of the 4 favorable outcomes with urological problems at birth, isolated hydronephrosis was found in one case, an abnormal renal cortical appearance in two cases, and, in the last case, hydronephrosis and an abnormal renal cortical appearance associated with oligohydramnios in late pregnancy was found at the end of the ultrasound monitoring of each pregnancy.

Concerning adverse outcome, of the 11 cases of IUFD, 8 had an abnormal kidney ultrasound appearance. Of the 50 TOP, 43 (86%) had an abnormal renal appearance on ultrasound. Of the 5 ND, all cases had an abnormal renal appearance.

Seventy-three karyotypes were determined (in 2 cases the parents refused) and nine of them (12.3%) were abnormal. No statistical association was found between bladder size and the risk of chromosomal abnormality.

In the 63 cases of adverse outcomes, the parents refused pathological examination of the fetus in 11 (17%) cases (9 because the karyotype was abnormal and 2 because the parents refused). Of the 52 (82.5%) cases where pathological examination was performed, 17 (32.7%) were not informative (11 by vacuum system and 6 by vaginal delivery). Finally, 35 were informative (67.3%) and megacystis was obstructive in 31 of these cases (88.5%), the majority of which (n = 19 or 54%) involved urethral stenosis or atresia, followed by posterior urethral valve, cloacal anomalies, and finally obstructive ureterocele. The origin of megacystis was non-obstructive in only 4 cases (11.4%). Details of fetal karyotypes and pathological characteristics are reported in Table 3.

## Discussion

In this observational study, our results show that fetuses with a bladder diameter of less than 12.5 mm can have a favorable outcome, with or without urological problems, with a high sensitivity and specificity (83.3 and 87.3%, respectively). Beyond this diameter, an adverse outcome is almost systematic.

With our results, 2 different situations can be highlighted. The first is where there is a high possibility of spontaneous regression when bladder size is less than 12.5 mm. A favorable outcome remains possible, albeit almost certainly with urological disease requiring short- or long-term care. In the case where diameter is greater than 12.5 mm, parents should be advised of certain adverse outcomes, like neonatal death or IUFD (Fig 3).

Our results are highly consistent with those of Kao et al, whose study was carried out at the same time as ours, showing that isolated megacystis < 12 mm is associated with a positive outcome [35]. Another strength of the current study is the size of the included population, which is one of the largest in the literature for a population concerning only megacystis in the first

**Table 3. Antenatal and postnatal characteristics of fetuses according to neonatal outcome.**

| COMPLEMENTARY EXAMS | All cases of megacystis | Bladder size | |
|---|---|---|---|
| | | < 12.5 mm | >12.5 mm |
| | n = 75 (100%) | n = 14 | n = 61 |
| **Karyotype** | | | |
| Normal | 64 (87.6) | 13 (10.9) | 44 (43.8) |
| Abnormal | 9 (12.3) | 1 (1.3) | 8 (8.2) |
| Trisomy 13 | 2 (2.7) | 0 | 2 (2.7) |
| Trisomy 18 | 5 (6.8) | 1 (1.3) | 4 (2.7) |
| Trisomy 21 | 1 (1.3) | 0 | 1 (1.3) |
| Other | 1 (1.3) | 0 | 1 (1.3) |
| Not wanted | 2 (2.7) | 1 (1.3) | 1 (1.3) |
| **Pathological diagnosis** | 52 (82) | Not done | 52 (65.4) |
| **Informative** | 35 (67.3) | | 35 (44.2) |
| **LUTO** | 31 (88.5) | | |
| Posterior urethral valves | 6 (17.1) | | 7 (9.6) |
| Urethral stenosis/urethral atresia | 19 (54) | | 17 (23.1) |
| Obstructing ureterocele | 2 (5.7) | | 3 (3.8) |
| Cloacal anomalies | 4 (11.4) | | 4 (3.8) |
| **Cause non-obstructive** | 4 (11.4) | | |
| Prune-Belly syndrome | 3 (8.6) | | 3 (1.9) |
| Other | 1 (2.8) | | 1 (1.9) |
| **Not wanted** | 11 (17.4) | | 11 (9.5) |
| **Uninformative** | 17 (32.7) | | 17 (21.1) |

(LUTO: lower urinary tract obstruction).

**Fig 3. Decision analysis and prenatal counseling according to the bladder size "cut-off" determined by the area under the ROC curve.** (TOP: termination of pregnancy; IUFD: in utero fetal death; ND: neonatal death).

trimester of pregnancy, with evaluation of ultrasound characteristics and neonatal outcomes [12, 28–30, 35]. Furthermore, ultrasound was carried out by trained practitioners in two diagnosis centers, allowing a reliable diagnosis of megacystis. Moreover, we were able to collect all pathological findings desired by the parents, even if some of them were not informative. Indeed, macroscopic examination of lower urinary tract obstruction is technically difficult due to the term and the size of fetus, with dissection not always possible, even if it alone can identify the type of barrier [36, 37].

Our study has several limitations. No multivariate analysis was performed. However, our study was essentially descriptive, and multivariate analysis would not have justified the analysis of all data. Furthermore, concerning follow-up of liveborn infants, there is no certainty about their long-term state of health, even if the follow-up at one year is reassuring. Indeed, the perspective on the state of health is different because our data collection is spread over ten years. Finally, although it is known as a prognostic factor [20, 38], the evaluation of renal function in the antenatal period could not be done because this analysis was not performed at the two university hospitals.

In accordance with previously published studies, our study confirms that fetuses with a large megacystis more frequently have an adverse outcome [21, 39]. It is important to adjust prenatal counseling of the parents in the case of early diagnosis. In fact, the most important questions are the possibility of regression, and neonatal outcomes if pregnancy is continued. Bladder diameter seems to be a predictive marker of neonatal outcome.

Several authors have been interested in spontaneous megacystis regression. Iuculano et al [28, 40] consider that an ultrasound scan performed 2 weeks after the megacystis diagnosis can predict the outcome in fetuses with a longitudinal bladder diameter < 15 mm as early as the end of the first trimester. In their study, the outcome of euploid fetuses with a longitudinal bladder diameter < 15 mm was favorable in 58.3% of cases. Fontanella et al reported in 2017 [19] that diameter bladder is a predictor of spontaneous resolution if the diagnosis is made before 18 weeks (80% sensitivity and 79% specificity). They specified that spontaneous regression before 23 weeks is a marker of favorable neonatal outcome, without urological surgical intervention, which is consistent with the 2017 study by Girard et al. [29].

Bladder size was not linked with chromosomal abnormality. However, we consider that karyotyping should be offered to parents, since a chromosomal abnormality can be found regardless of the size of the megacystis. There is therefore no link between bladder size and chromosomal abnormality. Concerning the final diagnosis by pathological examination of the fetus, we observed some disagreement with the literature data [41, 42]. In our population, the main diagnosis was urethral stenosis or atresia compared to PUV, a diagnosis frequently reported as the most important cause of megacystis in the second and third trimesters of pregnancy [41]. Two possibilities could explain this discrepancy. Firstly, the technical difficulties of pathological examination of the fetus could lead to underdiagnosis of PUV at this term. Secondly, there may be a tendency to superimpose the etiological diagnoses of megacystis whatever the trimester of pregnancy. But, etiologies of complete and total urinary tract obstruction (urethral atresia, urethral stenosis, and some cases of completely obstructive PUV) explain the early and strong ultrasound and clinical expression. Indeed, technical improvements in ultrasound equipment have ameliorated visualization of fetal anatomy in the first trimester.

The purpose of our study was mainly to improve prenatal counseling in the first trimester of pregnancy, by a response adapted to the first-trimester ultrasound findings. The proposal for a prognosis threshold based on neonatal outcome seems appropriate and necessary, in a society where end-of-life support is increasingly important.

## Conclusion

Two different situations with different neonatal outcomes can be highlighted with megacystis in the first trimester of pregnancy. Fetuses with megacystis < 12.5 mm have a significantly higher chance of a favorable outcome compared to megacystis > 12.5 mm. Bladder diameter appears to be a predictive marker of neonatal outcome.

Similarly, it seems better to speak of low urinary tract obstacle rather than of PUV in cases of first-trimester megacystis. Urethral stenosis and urethral atresia are the commonest diagnoses. Regardless of bladder diameter, karyotyping remains important. Optimized first-trimester screening is now an integral part of our screening policy. Taken together, these investigations improve prenatal counseling by providing an adapted and adjusted response to the first-trimester ultrasound findings.

## Supporting information

**S1 Data.**
(XLSX)

## Author Contributions

**Conceptualization:** Emmanuelle Lesieur, Mathilde Barrois.

**Formal analysis:** Mathilde Bourdon.

**Investigation:** Emmanuelle Lesieur, Laurence Loeuillet, Clémence Delteil, Julia Torrents.

**Methodology:** Emmanuelle Lesieur, Mathilde Barrois, Mathilde Bourdon, Olivia Anselem.

**Supervision:** Vassilis Tsatsaris, Olivia Anselem.

**Validation:** Emmanuelle Lesieur, Julie Blanc, Florence Bretelle, Gilles Grangé, Vassilis Tsatsaris, Olivia Anselem.

**Visualization:** Emmanuelle Lesieur, Florence Bretelle, Gilles Grangé, Vassilis Tsatsaris, Olivia Anselem.

**Writing – original draft:** Emmanuelle Lesieur.

**Writing – review & editing:** Emmanuelle Lesieur.

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
