## [Decision Letter · Decision Letter 0]

8 Feb 2021

PONE-D-20-39804

MEGACYSTIS IN THE FIRST TRIMESTER OF PREGNANCY: PROGNOSTIC FACTORS AND PERINATAL OUTCOMES

PLOS ONE

Dear Dr. Lesieur,

Thank you for submitting your manuscript to PLOS ONE. After careful consideration, we feel that it has merit but does not fully meet PLOS ONE’s publication criteria as it currently stands. Therefore, we invite you to submit a revised version of the manuscript that addresses the points raised during the review process.

We look forward to receiving your revised manuscript.

Kind regards,

Rogelio Cruz-Martinez, Ph.D.

Academic Editor

PLOS ONE

Journal Requirements:

2. You indicated that you had ethical approval for your study.

In your Methods section, please ensure you have also stated whether you obtained consent from parents or guardians of the minors included in the study (if applicable) or whether the research ethics committee or IRB specifically waived the need for their consent.

3. In the ethics statement in the manuscript and in the online submission form, please provide additional information about the patient records/samples used in your retrospective study.

Specifically, please ensure that you have discussed whether all data/samples were fully anonymized before you accessed them and/or whether the IRB or ethics committee waived the requirement for informed consent.

If patients provided informed written consent to have data/samples from their medical records used in research, please include this information.

Additional Editor Comments:

The authors included a large series of first trimester fetuses with megacystis and described the neonatal outcomes according to the bladder size.

I agree with one of the reviewers that this large series merits publication on this Journal but I also agree with the main criticism by the second reviewer that bladder size groups were arbitrarily selected and that fetal bladder size should be consider a continuous variable and therefore I strongly recommend to assess the clinical value of bladder size to predict neonatal outcome by a decision tree analysis, which allow an automatic classification using the predictive variable as a continuous variable allowing a sequential analysis to predict postnatal outcome. Since previous studies have demonstrated that the size of fetal bladder predicts adverse neonatal outcome, I think that assessing an automatic fetal bladder size cut-off may be of clinical interest and thus, deserving to be publish in this Journal

Please include the following references:

Fontanella et al. Antenatal Workup of Early Megacystis and Selection of Candidates for Fetal Therapy. Fetal Diagn Ther 2019;45(3):155-161.

Reviewers' comments:

Reviewer's Responses to Questions

**Comments to the Author**

1. Is the manuscript technically sound, and do the data support the conclusions?

Reviewer #1: Yes

Reviewer #2: No

2. Has the statistical analysis been performed appropriately and rigorously? 

Reviewer #1: Yes

Reviewer #2: No

3. Have the authors made all data underlying the findings in their manuscript fully available?

Reviewer #1: Yes

Reviewer #2: No

4. Is the manuscript presented in an intelligible fashion and written in standard English?

Reviewer #1: Yes

Reviewer #2: No

5. Review Comments to the Author

Reviewer #1: The authors report a multicenter cohort study, which evaluates the prognosis associated with the bladder size at 1st trimester in case of megacystis.

This large and interesting series deserve to be published. However it should be more focus on its originality that lies in the analysis of outcomes based on bladder size. Moreover, data on evolution during pregnancies and renal function should be provided.

INTRODUCTION

L66-68: The sentence is a bit unclear. Do the authors mean ‘after’ rather ‘with’ checking of bladder emptying during the exam’ ?

L73-78: The authors should rather justify the interest of the present study by describing the lack of data regarding the prognosis associated with bladder size, so as the etiology depending on the bladder size.

L78-80: The main objective should be rephrased to: ‘…in case of megacystis at first trimester, whether bladder size is associated to unfavorable outcome’

L80-83: I suggest to focus only on etiology based on bladder size (rather than to also aiming to describe the whole series as an objective)

METHODS

L95-100 : The authors should justify further the choice of these thresholds.

L103-105: It would be very important to identify cases with or without normal renal function among live births if there is a one-year follow-up+++ Dead/Alive is not appropriate for a study on megacystis and renal function is of interest.

L109: ‘all patients’ rather than ‘all fetuses’

L116: There was no karyotypes performed in one of the centers???

L112-115: Authors should precisely describe the parameters related to the inclusion criteria. How was measured the bladder? Which axis? What if the measurement is different at two different time points during the US examination or between 2 US?

RESULTS

Again, reporting renal function would be really important.

L176-177: Authors introduce here the evolution of US findings at second US. Is it second trimester US? It would be of interest indeed to have follow-up data but it should be more detailed (including bladder aspect, gestational age,…) in a specific table.

L184-186: Mixing data from both US is a bit confusing. If inclusion so as tables based on first US, it should be clearly stated in the methods section so as in the results section.

L191-216: This section is too long, a bit hard to understand, and not necessary. First information is already provided in Table 3 and above all the article should focus on the outcome based on the bladder size.

DISCUSSION

Discussion is interesting and figure 2 is appropriate.

Reviewer #2: This appears to be a rather large series of a rare fetal condition.

However, megacystis was only transient in a large proportion with good outcomes as expected

The authors arbitrarily grouped cases by bladder size. There is no rationale for this and bladder size should therefore be considered a continuous variable

There are no information on standardized care pathway or on indications for TOPs

The etiologies for LUTO in this series are similar to those reported in previous studies

The conclusions are trivial and not substantiated by the data, as explained above.

6. PLOS authors have the option to publish the peer review history of their article (what does this mean?). If published, this will include your full peer review and any attached files.

Reviewer #1: No

Reviewer #2: No

---

## [Author Response · Author response to Decision Letter 0]

18 Mar 2021

To Plos One Editorial

The 23rd of february 2021

Thank you very much for reviewing our manuscript “Megacystis in the first trimester of pregnancy: prognostic factors and perinatal outcomes”. We also greatly appreciate the reviewers for their complimentary comments and suggestions. We have carried out the experiments that the reviewers suggested and we revised the manuscript accordingly.

We hope that you find our responses satisfactory and that the manuscript meets Plos One publication criteria as it currently stands.

Sincerely,

Dr LESIEUR Emmanuelle

---

## [Editor Report · Decision Letter 1]

30 Mar 2021

PONE-D-20-39804R1

MEGACYSTIS IN THE FIRST TRIMESTER OF PREGNANCY: PROGNOSTIC FACTORS AND PERINATAL OUTCOMES

PLOS ONE

Dear Dr. Lesieur,

Thank you for submitting your manuscript to PLOS ONE. After careful consideration, we feel that it has merit but does not fully meet PLOS ONE’s publication criteria as it currently stands. Therefore, we invite you to submit a revised version of the manuscript that addresses the points raised during the review process.

The manuscript has been improved according to the reviewers' comments. However, a main concern persists throughout the manuscript, mainly due to the use of pre-defined bladder size groups instead to select an automatic bladder size cut-off to predict prognosis. 

We look forward to receiving your revised manuscript.

Kind regards,

Rogelio Cruz-Martinez, Ph.D.

Academic Editor

PLOS ONE

Editor comments:

The authors should grouped the prognostic category based in the analysis on bladder size and not in a pre-defined bladder size groups, i.e. to analyze bladder size as a continuous variable and selecting the best cut-off to predict adverse outcome. The mean bladder size was significantly higher in cases with adverse outcome, but it remains unknown which is the best cut-off to predict such adverse outcome.

---

## [Author Response · Author response to Decision Letter 1]

16 Apr 2021

Thank you very much for reviewing again our manuscript “Megacystis in the first trimester of pregnancy: prognostic factors and perinatal outcomes”. We also greatly appreciate the editor for his complimentary comment.

Editor comments :

The authors should grouped the prognostic category based in the analysis on bladder size and not in a pre-defined bladder size groups, i.e. to analyze bladder size as a continuous variable and selecting the best cut-off to predict adverse outcome. The mean bladder size was significantly higher in cases with adverse outcome, but it remains unknown which is the best cut-off to predict such adverse outcome.

Our response : 

As recommended by this one, we carried out a new statistical analysis using a ROC curve to determine a “bladder diameter cut – off”, making it possible to determine a diameter for which we can predict the neonatal outcome with a high sensitivity and specificity.

Based on the ROC curve analysis, the optimal “cut-off” bladder diameter was 12,5 mm. Under this cut-off, the probability to have a favorable outcome was 91 %, with sensitivity 83,3 % CI 95 (33,3 – 91,7 %) and specificity was 87,3 % CI 95 (75 – 99,9)

Two groups were then compared : (i) bladder diameter under 12,5 mm; (ii) bladder diameter greater than 12,5 mm. Our population was distributed according to this cut off. 

All tables and figures have been changed. A new figure, with the ROC curve was created.

Big changes have been made to the manuscript. 

We hope that you find our responses satisfactory and that the manuscript meets Plos One publication criteria as it currently stands.

---

## [Editor Report · Decision Letter 2]

21 May 2021

PONE-D-20-39804R2

MEGACYSTIS IN THE FIRST TRIMESTER OF PREGNANCY: PROGNOSTIC FACTORS AND PERINATAL OUTCOMES

PLOS ONE

Dear Dr. Lesieur,

Thank you for submitting your manuscript to PLOS ONE. After careful consideration, we feel that it has merit but does not fully meet PLOS ONE’s publication criteria as it currently stands. Therefore, we invite you to submit a revised version of the manuscript that addresses the points raised during the review process.

The manuscript has been improved according to the editor and reviewer's suggestions but I still consider it is not suitable for publication in its current form. 

We look forward to receiving your revised manuscript.

Kind regards,

Rogelio Cruz-Martinez, Ph.D.

Academic Editor

PLOS ONE

Journal Requirements:

Additional Editor Comments (if provided):

The manuscript has been improved according to the editor and reviewer's suggestions and I consider it is now suitable for publication in its current form

Minor comments:

The manuscript still require a gramatical correction.

Please rephrase the following sentence in the abstract in order to clarify the meaning "Pathological examination of the fetus was performed in 52 (82.5%)".

Please define "neonatal outcome" in the Abstract.

In the Methods section, neonatal outcome has been defined as a live birth with or without normal renal function. Such outcome should be defined as "neonatal survival"

The clinical algorithm should include prediction of neonatal survival as a primary outcome and prediction of renal failure as a secondary outcome

Please clarify the meaning of the following sentence "In the case of TPO; pathological examination of the fetus was suggested". Does it mean fetal autopsy?

---

## [Author Response · Author response to Decision Letter 2]

25 May 2021

Dear Editor, Dear reviewers

I am pleased to submit again our revised manuscript “Megacystis in the first trimester of pregnancy : prognostic factors and perinatal outcomes” by E. Lesieur, MD, M. Barrois, MD, M. Bourdon, MD, J. Blanc, MD, L Loeuillet, MD, C. Delteil, MD, J. Torrents, MD, F. Bretelle, MD PhD, G. Grangé, MD, V. Tsatsaris, MD PhD, O. Anselem, MD

We also greatly appreciate the editor complimentary comments and journal requirements. 

We revised the manuscript accordingly.

We hope that you find our responses satisfactory and that the manuscript meets Plos One publication criteria as it currently stands.

Sincerely,

Dr LESIEUR Emmanuelle

---

## [Editor Report · Decision Letter 3]

15 Jun 2021

PONE-D-20-39804R3

MEGACYSTIS IN THE FIRST TRIMESTER OF PREGNANCY: PROGNOSTIC FACTORS AND PERINATAL OUTCOMES

PLOS ONE

Dear Dr. Lesieur,

Thank you for submitting your manuscript to PLOS ONE. After careful consideration, we feel that it has merit but does not fully meet PLOS ONE’s publication criteria as it currently stands. Therefore, we invite you to submit a revised version of the manuscript that addresses the points raised during the review process.

We look forward to receiving your revised manuscript.

Kind regards,

Rogelio Cruz-Martinez, Ph.D.

Academic Editor

PLOS ONE

Journal Requirements:

Additional Editor Comments (if provided):

The manuscript has been improved accordingly. However, I strongly recommend to have a grammar correction by a native English speaker because there are still several typographical errors.

---

## [Author Response · Author response to Decision Letter 3]

19 Jun 2021

Thank you very much for reviewing again our manuscript “Megacystis in the first trimester of pregnancy: prognostic factors and perinatal outcomes”. We also greatly appreciate the editor for his complimentary comment.

Journal requirements :

Please review your reference list to ensure that it is complete and correct.

All other references are correct and correspond to the manuscript. The reference list is complete. 

Editor comments :

- The manuscript has been improved accordingly. However, I strongly recommend to have a grammar correction by a native English speaker because there are still several typographical errors.

We agree with the reviewer’s comment. Grammar correction has been made by a native English speaker. 

We hope that you find our responses satisfactory and that the manuscript meets Plos One publication criteria as it currently stands.

Sincerely,

Dr LESIEUR Emmanuelle

---

## [Editor Report · Decision Letter 4]

27 Jul 2021

MEGACYSTIS IN THE FIRST TRIMESTER OF PREGNANCY: PROGNOSTIC FACTORS AND PERINATAL OUTCOMES

PONE-D-20-39804R4

Dear Dr. Lesieur,

We’re pleased to inform you that your manuscript has been judged scientifically suitable for publication and will be formally accepted for publication once it meets all outstanding technical requirements.

Kind regards,

Rogelio Cruz-Martinez, Ph.D.

Academic Editor

PLOS ONE

---

## [Editor Report · Acceptance letter]

27 Aug 2021

PONE-D-20-39804R4 

Megacystis in the first trimester of pregnancy: Prognostic factors and perinatal outcomes 

Dear Dr. Lesieur:

I'm pleased to inform you that your manuscript has been deemed suitable for publication in PLOS ONE. Congratulations! Your manuscript is now with our production department. 

Kind regards, 

on behalf of

Dr Rogelio Cruz-Martinez 

Academic Editor

PLOS ONE